# Molecular Epidemiology of *Brucella* spp. in Aborted Livestock in the Ningxia Hui Autonomous Region, China

**DOI:** 10.3390/vetsci12080702

**Published:** 2025-07-28

**Authors:** Cai Yin, Cong Yang, Yawen Wu, Jing Di, Taotao Bai, Yumei Wang, Yuling Zhang, Longlong Luo, Shuang Zhou, Long Ma, Xiaoliang Wang, Qiaoying Zeng, Zhixin Li

**Affiliations:** 1College of Veterinary Medicine, Gansu Agricultural University, Lanzhou 730070, China; yincai22@163.com (C.Y.); 17711805237@163.com (J.D.); 2Ningxia Hui Autonomous Region Center for Animal Disease Control and Prevention, Yinchuan 750011, China; wuyawen_123@163.com (Y.W.); baitao198487@163.com (T.B.); ruirui86526@sina.com (Y.W.); zhangyuling-1001@163.com (Y.Z.); 13639583806@163.com (L.L.); zhou_shuang221@163.com (S.Z.); malong20185561@163.com (L.M.); nxwxl-1019@163.com (X.W.); 3Ningxia Hui Autonomous Region Center for Disease Control and Prevention, Yinchuan 750001, China; yangcong0829@163.com; 4School of Agriculture, Ningxia University, Yinchuan 750021, China

**Keywords:** abortion, livestock, zoonosis, *Brucella* spp., molecular identification, genotype

## Abstract

Brucellosis is a bacterial infection caused by the *Brucella* species that leads to a series of symptoms ranging from fever to abortion both in livestock and humans. It severely impacts the development of the livestock industry and public health. The human Brucellosis epidemic in Ningxia has gradually worsened in recent years. To reduce the impact of this epidemic and the effect of Brucellosis on livestock, the “Ningxia Brucellosis Prevention and Control Special Three-Year Action Implementation Plan (2022–2024)” was implemented by the government of Ningxia. The plan includes information regarding the surveillance, epidemiological investigations, and ongoing compulsory immunization of newborn livestock. As a part of the surveillance and epidemiological investigation, this study comprises the first systematic study on *Brucella* spp. from the aborted tissues of livestock in Ningxia, China. In this cross-sectional study, which was conducted in Ningxia from 2022 to 2023, we reported that 8.68% of the aborted tissue samples of livestock were identified as being positive for *Brucella* spp. This indicates a high risk of transmission from the aborted livestock to other animals and humans. Therefore, comprehensive disinfection and safe treatment should be implemented after the abortion of livestock to interrupt the transmission pathway in order to prevent the further infection of livestock and humans.

## 1. Introduction

Abortion represents a serious issue in pregnant animals, leading to significant financial challenges, especially in developing nations [1]. This phenomenon not only limits production but also severely impacts food production and the agricultural industry. Furthermore, potential zoonotic pathogens related to abortion may pose serious threats to public health. Identifying the causal agent responsible for an abortion event is significant for effective disease control and management in order to address the public health threats that are associated with zoonotic diseases, in addition to preventing further outbreaks [2].

Brucellosis is a highly prevalent zoonotic disease worldwide. Globally, the annual number of infections in humans is approximately 2.1 million [3]. This systemic zoonotic infection is manifested as a series of symptoms ranging from fever to miscarriage. Various species within the *Brucella* genus are responsible for Brucellosis, which results in substantial economic losses and considerable public health challenges [4]. It is a major contributor to infectious abortion in livestock across various nations [5] and presents a grave occupational risk for individuals such as slaughterhouse workers, farmers, shepherds or traders, veterinarians, and lab technicians [6].

*Brucella* spp. live within cells, serving as abortive agents in various animal species, particularly affecting cattle, goats, sheep, swine, and dogs [7]. The bacteria can enter hosts through ingestion, the inhalation of air or dust, or through the conjunctiva or damaged skin [8]. The *Brucella* genus comprises several species, with *B. abortus*, *B. melitensis*, and *B. suis* being found most frequently in domesticated animals [9]. *B. melitensis* is responsible for the majority of Brucellosis cases in sheep and goats and can cause abortions in infected pregnant animals [10]. Human forms of Brucellosis are most commonly contracted from cattle, sheep, goats, camels, and pigs [11]. Typically, humans contract the infection via direct or indirect exposure to infected animals or contaminated materials, such as tissues from aborted livestock [12]. Numerous zoonotic species of *Brucella* are obtained from animals that have experienced abortions.

Brucellosis is commonly found in China, particularly in the northern regions, where the local economy relies heavily on ruminant animals [13], and *B. melitensis* ST8 is the most common species, with it being widespread in 1990–2010 [14]. The Ningxia Hui Autonomous Region (104°17′–107°39′ E, 35°14′–39°23′ N) in northwest China has a dry climate and sufficient daylight. It has a long history in relation to the development of the livestock industry. By the end of 2022, the number of dairy cow, beef cattle, and Tan sheep herds in the region reached 0.837 million, 1.484 million, and 7.105 million, respectively (http://nxccpit.nx.gov.cn/xwzx/zzqxx/202304/t20230426_4047068.html (accessed on 4 May 2025)). Due to the rapid expansion of animal farming and frequent animal movement, the prevalence of bovine Brucellosis in the area was 2.46% and that of sheep was 3.85% in 2022. The human Brucellosis epidemic has gradually worsened, with the incidence increasing from 0.441/100,000 in 2005 to 86.83/100,000 in 2022 in Ningxia [15].

In order to systematically investigate the presence of Brucellosis in aborted livestock in Ningxia, China, and analyze the genetic characteristics of the isolated *Brucella* spp., bioinformatics techniques such as Multilocus Sequence Typing (MLST) and multiple-locus variable-number tandem repeat analysis (MLVA) genotyping were employed to perform analyses. In this study, we aim to clarify its molecular evolution and improve prevention and control strategies.

## 2. Materials and Methods

### 2.1. Sample Collection

The following study was a cross-sectional study performed from January 2022 to December 2023. The study population comprised livestock that had experienced abortions in Ningxia, China. Veterinarians from various districts and areas were informed about the study’s aim and received training on biosafety procedures. Where possible, samples of aborted placenta or fetal spleen were taken from farms reporting livestock that had experienced abortions. The samples were collected into 50 mL sterile conical centrifuge tubes. These tubes were sealed twice in biohazard plastic bags by veterinarians who were equipped with appropriate personal protective equipment [1]. The cold chain was maintained, with samples being transported on ice and stored at −20 °C before analysis. After collecting the samples, thorough disinfection and safe treatment procedures were performed.

### 2.2. Isolation and Culture of Brucella spp. from Aborted Samples

*Brucella* spp. were isolated and cultured in a BSL-3 laboratory of the Ningxia Hui Autonomous Region Center for Disease Control and Prevention and subsequently tested following a previously described method [16,17]. Briefly, tissue homogenates were inoculated onto *Brucella* Selective Medium plates with the following antibiotics: bacitracin (25,000 IU/L), mycostatin (10,000 IU/L), polymyxin B (5000 IU/L), vancomycin (20 mg/L), and actinone (10 mg/L). These plates were incubated in the presence of 5% CO_2_ at 37 °C and subsequently examined every day for 3 weeks. Suspected colonies were preliminarily identified using biochemical tests, such as urease production and carbon dioxide requirement and Gram and Ziehl-Neelsen staining. Suspected colonies were purified, and agglutination was observed with monospecific antisera A, M, and R for strain typing and biotyping.

### 2.3. Molecular Analysis

#### 2.3.1. DNA Extraction

Bacteria present in the samples were inactivated through heating at 100 °C for 5 min. A Nucleic Acid Automatic Extraction System (Xi’an Tianlong Technology Co., Ltd., Xi’an, China) was then employed to extract the bacterial genomic DNA from the cultures and tissues according to the supplier’s protocols. The extracted DNA was maintained at −20 °C until use.

#### 2.3.2. Detection of *Brucella* spp. via qPCR

Primers for *bcsp31* were designed using Primer Premier 6 (Premier Biosoft, San Francisco, CA, USA). Forward Primer: 5′-GGCCATCTCGAACGGTATTT-3′, Reverse Primer: 5′-GTTTCCGCCACGTCCTTATAG-3′, probe: 5′-(FAM, 6-carboxyfluorescein) CTTCTTCTCGAAGGATGTGGTTCCTGC-BHQ1 (Black Hole Quencher 1)-3′. Sangon Biotech (Shanghai) Co., Ltd. (Shanghai, China) synthesized the primers and probe.

The qPCR 25 µL reaction comprised 12.5 µL of Premix Ex Taq (Takara Bio Inc., Dalian, China), 0.5 µL of each primer (final concentration = 0.2 μM), 1 µL of each probe (final concentration = 0.2 μM), 2 µL of DNA template, and 8.5 μL of ddH_2_O. The source of DNA for qPCR was tissues and culture isolation. The qPCR analysis was performed on a CFX96 Real-Time PCR Detection System (Bio-Rad, Hercules, CA, USA) using cycling comprising initial denaturation at 95 °C for 20 s and then 40 cycles of denaturation at 95 °C for 5 s, before annealing/extension at 60 °C for 20 s. The FAM channel was chosen to monitor the fluorescence signal at 60 °C.

#### 2.3.3. Identification of *Brucella* spp.

The vaccine strains (S2 and A19) of *Brucella* spp. were identified using an identification kit (Aodong Inspection & Testing Co., Ltd., Shenzhen, China).

*B. abortus* biovar (bv). 1, 2, and 4; *B. melitensis* bv. 1, 2, and 3; *B. ovis;* and *B. suis* bv. 1 were identified using a multiplex AMOS-PCR assay (named after the species detected: *B. abortus*, *B. melitensis*, *B. ovis*, and *B. suis*) [18]. The positive controls comprised *B. suis*, *B. melitensis*, *and B. abortus* nucleic acids (Pudao Standard Technology Co., Ltd., Beijing, China). A PCR Instrument (Mastercycler Nexus GX2, Eppendorf, Hamburg, Germany) was used to perform the amplification reactions. The reaction (25 μL) contained 1 × MyRaq Red PCR Mix, four species-specific forward primers and reverse primer IS711 (final concentrations = 0.1 μM and 0.5 μM, respectively), and 2 μL of DNA template. The PCR cycling conditions comprised 95 °C for 3 min; 40 cycles of 95 °C for 1 min, 60 °C for 2 min, and 72 °C for 1 min; and lastly 72 °C for 10 min. UV illumination was used to visualize amplicons after electrophoretic separation on a 1.5% agarose gel stained with GelRed.

#### 2.3.4. MLST Genotyping

Nine separate genomic loci were selected for MLST genotyping, comprising seven housekeeping genes (*cobQ*, *trpE*, *gyrB*, *dnaK*, *glk*, *aroA*, and *gap*), one intergenic fragment (int-hyp), and one outer membrane protein gene (*omp25*) [19]. The PCR reactions were carried out in 50 μL volumes, and 5 μL of each reaction for each of the nine loci was subjected to capillary electrophoresis by Sangon Biotech (Shanghai) Co., Ltd. A web-based MLST service (*Brucella* Base, http://59.99.226.203/brucellabase/mlst.html (accessed on 4 May 2025)) was used to predict each sequence type (ST) over all loci.

To further investigate the molecular epidemiology of *Brucella* spp. strains at the national and international levels, data from previous studies listed in public databases for molecular typing and microbial genome diversity (https://pubmlst.org/organisms/brucella-spp (accessed on 4 May 2025)) were employed. PHYLOViZ 2.0 software [20] was used to construct a minimum spanning tree (MST) to evaluate the genetic relationships among the 14 isolates.

#### 2.3.5. MLVA Genotyping

MLVA analysis was conducted using a previously described method [21]. Briefly, 16 primer pairs were used, grouped into three panels: Panel 1 (bruce06, bruce08, bruce11, bruce12, bruce42, bruce43, bruce45, and bruce55); Panel 2A (bruce18, bruce19, and bruce21); and Panel 2B (bruce04, bruce07, bruce09, bruce16, and bruce30). The PCR amplifications were performed in a 50 μL reaction volume, 5 μL of which was subjected to analysis using 2% gel electrophoresis (with gel red nucleic acid staining), followed by UV visualization and photography. BioNumerics version 5.1 (Applied Maths NV, Sint-Martens-Latem, Belgium) was employed to estimate the band densities. The allele numbering system [22] was then used to convert the band densities into repeat units. BioNumerics version 5.1 was then used to generate arithmetic averages for unweighted pair group phylogenetic analysis of the 14 strains isolated from aborted livestock in the study and 21 strains isolated by Liu et al. from humans [15].

### 2.4. Analysis of the Data

We calculated the proportion of positive animals by dividing the total number of animals that tested positive by the total number of animals in the study. Microsoft Excel 2010 spreadsheets (Microsoft Corp., Redmond, WA, USA) were employed to record the data. Proportions were calculated, and positive results were assessed using the *χ*^2^ test in Epi-Info 7 version 10 (CDC, Atlanta, GA, USA).

## 3. Results

### 3.1. Detecting Brucella spp. in Cattle and Sheep Abortion Tissues

In total, 831 cases of abortion were reported during the study period, and 749 samples of aborted placenta or fetal spleen were collected from the aborted livestock (Figure 1); the remaining 82 cases were disposed of by the farmers themselves and were not sampled. The samples were obtained from 121 (16.15%) dairy cows, 94 (12.55%) beef cattle, and 534 (71.29%) sheep. Among them, 14.95% (112/749) originated from Shizuishan City, 20.43% (153/749) originated from Yinchuan City, 22.03% (165/749) originated Wuzhong City, 12.68% (95/749) originated from Zhongwei City, and 29.91% (224/749) originated from Guyuan City (Figure 2).

In this study, 8.68% (65/749) of the samples were identified as positive for *Brucella* DNA using qPCR. The PCR-positive rate of aborted tissues from beef cattle was 13.83% (13/94); for dairy cows, the rate was 9.09% (11/121), which was higher than the rate of 7.68% (41/534) for sheep. In total, 14 strains of *Brucella* spp. were isolated.

### 3.2. Identification of Brucella spp.

Amplicons of 731 bp (*B. melitensis*), 498 bp (*B. abortus*), and 285 bp (*B. suis*) were detected using AMOS-PCR in *Brucella* cultures from aborted cattle and sheep tissues. Among the 14 wild strains, 11 were recognized as *B. melitensis*, 2 were recognized as *B. abortus*, and 1 was recognized as *B. suis*.

### 3.3. MLST Genotyping

Using the nine MLST loci, four known STs were identified: ST2 (2-1-2-2-1-3-1-1-1; *n* = 2), ST7 (3-5-3-2-1-5-3-2-10; *n* = 2), ST8 (3-2-3-2-1-5-3-2-8; *n* = 9), and ST14 (1-6-4-1-4-3-5-1-2; *n* = 1). The predominant ST in the Ningxia region was ST8 (Figure 3).

### 3.4. MLVA Genotyping

MLVA was employed to characterize 14 strains isolated from aborted livestock. The unweighted pair group method with arithmetic averages (UPGMA) was used to cluster the strains based on their genetic similarities. A total of 21 strains isolated from humans, reported in the study by Liu et al., from Ningxia in 2019 were selected and analyzed together [15]. Panel 1 shows that the strains could be clustered into five MLVA-8 genotypes (42 (1-5-3-13-2-2-3-2; *n* = 29), 43 (1-5-3-13-2-3-3-2; *n* = 2), 47 (3-4-2-13-4-2-3-3; *n* = 1), 112 (1-5-3-13-2-3-3-2; *n* = 2), and 6 (2-3-6-10-4-1-5-2; *n* = 1)). Seven MLVA-11 genotypes were identified (GT116, GT108, GT125, GT106, GT210, GT136, and GT33), with the MLVA-11 genotype 116 representing the predominant genotype circulating in both humans and livestock, which is widely distributed throughout Ningxia.

Ten samples contained shared genotypes (NXhuman7 and NXhuman21, NXhuman12 and NXsheep4, NXhuman3 and NXsheep10, NXhuman5 and NXsheep9, NXhuman4, and NXbovine1), eight of which shared genotypes in humans and livestock. The other 10 strains isolated from aborted livestock and 15 strains isolated from humans were represented by unique strains (Figure 4).

## 4. Discussion

Various factors can contribute to pregnancy loss, including issues of a noninfectious nature such as nutritional deficiencies and physical stress [23]. In addition, infectious agents, such as viruses, bacteria, fungi, and protozoa, can also cause abortions. It is estimated that infectious pathogens are responsible for roughly 90 percent of abortion cases, with the main factors responsible being *Brucella* spp., *Chlamydia abortus*, *Toxoplasma gondii*, bovine viral diarrhea virus (BVDV), and so forth [24]. Notably, the genus *Brucella* spp. is one of the primary bacterial agents that causes abortions in livestock across numerous developing nations [1].

Brucellosis can lead to a series of symptoms, such as miscarriage and orchitis, both in animals and humans, with it having significant global economic impacts related to animal health and financial losses, which also threaten human health [8]. In India, Brucellosis has led to financial losses estimated at USD 3.4 billion, with an average loss of roughly 18.2 dollars for each buffalo, 6.8 dollars for each cow, 0.7 dollars for sheep, 0.6 dollars for pigs, and 0.5 dollars for goats [25]. Furthermore, bovine brucellosis has caused economic damages of up to 600 million dollars in Latin America [26]. There has been a steady rise in human Brucellosis cases from the year 2000, affecting all of mainland China. The number of human cases has grown from 18,416 in 2005 to 75,858 in 2023 (https://www.ndcpa.gov.cn/jbkzzx/c100016/common/list.html (accessed on 4 May 2025)). In Ningxia, the number of human cases increased from 26 in 2005 to 6292 in 2022 [15]. The most effective method to prevent human Brucellosis cases is to control the disease in livestock. Effective strategies, such as regular livestock screening, removal of infected groups, and immunizing healthy animals, have led to the successful management of the disease in numerous developed countries [12].

To reduce the impact of Brucellosis on livestock, promote further development of the livestock industry, and ensure public health security, the “Ningxia Brucellosis Prevention and Control Special Three-Year Action Implementation Plan (2022–2024)” was implemented by the government of Ningxia [27]. The Plan encompasses a series of systematic measures aimed at both humans and livestock. These measures include the ongoing compulsory immunization of newborn lambs with the S2 vaccine and calves with the A19 vaccine and comprehensive disinfection and safe treatment of aborted livestock. In addition, the Plan emphasizes surveillance and epidemiological investigations, enhanced public awareness campaigns, and the training of both professional and reserve personnel. Furthermore, it highlights the implementation of a special “mask and gloves initiative” to ensure comprehensive safety. The initiative provides free gloves and masks to workers in high-risk industries, such as breeding and slaughtering. Vaccination can effectively decrease the incidence of Brucellosis in livestock and correlates with a reduced number of reported cases in humans, despite there currently being no human vaccines available [28]. Importantly, reducing infectious disease spread is supported by appropriate protective measures [27]. Employees involved in breeding, transport, slaughtering, and processing have seen a significant increase in their awareness of zoonotic diseases such as Brucellosis, tuberculosis, and anthrax prevention and control. Furthermore, public awareness of epidemic prevention and the control of zoonotic diseases have been continuously improved through the implementation of systematic measures. The number of reported human Brucellosis cases in the entire region has decreased by 28.5% year-on-year since 2023.

As a part of surveillance and epidemiological investigations, this study is a vital element of the Plan to reduce reported human cases, comprising the first systematic study on *Brucella* spp. from aborted tissues of livestock in the Ningxia Hui Autonomous Region of China. Our results indicate that in Ningxia, at least three *Brucella* species (*B. suis*, *B. abortus*, and, *B. melitensis*) are circulating, among which *B. melitensis* ST8 was the most frequently detected species.

In specific areas of infection, determining effective treatment strategies, tracing transmission, clarifying the reservoir and source, evaluating epidemiological intensity, and classifying foci are crucially informed by the identification of *Brucella* genotype, biovars, and species [9]. Correlations among *Brucella* spp. cannot be analyzed using classical bio-typing methods; however, tracking infection spread, genetic diversity, and population structure can be aided by molecular epidemiological studies, thereby facilitating and improving effective measures for disease control and prevention [29]. Moreover, *Brucella* MLST and MLVA have been suggested as supplemental methodologies to traditional bio-typing techniques [13].

The main *Brucella* spp. in China include the following: *B. melitensis* biovar 1 and 3 (such as ST7, ST8, ST34, ST35, and ST37), *B. abortus* biovar 1 and 3 (such as ST1, ST2, ST5, ST28, ST29, ST30, ST31, ST32, ST33, and ST38), and *B. suis* biovar 1 and 3 (such as ST14, ST17 and ST36), with *B. melitensis* as the main epidemic pathogenic species [14]. *B. melitensis* ST8 also constitutes the main species in Inner Mongolia [30], Xinjiang [24], Qinghai [31], and Guangxi [32], with it being responsible for numerous cases of human Brucellosis. This finding is also supported by the detection of 11 *B. melitensis* strains, and 9 ST8 among the 14 isolated strains in this study. *B. melitensis* ST8’s prevalence in Northwest China, and even in southern China more recently, may reflect the regional fitness advantages of the strain. Its prevalence has mainly resulted from the frequent transportation and circulation of live livestock on the Chinese mainland.

Although MLST and MLVA showed similarly consistent discrimination at the *Brucella* species level, MLVA could discriminate the genotype with higher resolution than MLST [13]. A complete MLVA-16 assay is often employed to traceback infection sources and investigate molecular epidemiological associations among *Brucella* isolates [30]. In the study presented herein, there eight strains shared the same MLVA-16 genotypes in humans and livestock. This finding demonstrates that the human and livestock cases were caused by a common source. Other strains were represented by unique strains. Brucellosis is primarily transmitted via direct or indirect contact between infected animals or products and humans [12], which indicates that some human cases cannot be traced to animal sources and that the distribution of *Brucella* species among domestic animals is in fact more diverse.

Bacterial isolation is considered indisputable evidence of disease [12]. However, some samples (aborted fetuses and vaginal swabs) assessed as positive based on PCR results but negative based on culture likely occurred as a result of contamination [24] and also arose from an infection that had run its course, with no live bacteria present but instead residual debris from dead cell’s DNA. In addition, there can also be a gap of several days between the time of reporting abortion and the time of sampling; in particular, at higher temperatures, old and decaying aborted tissues may also decrease the isolation rate of the bacteria.

This study suffers from some inherent limitations. Firstly, the submitted number of abortion cases may not provide an accurate representation of the actual number of cases. The underreporting of livestock abortions represents a global issue, even when reporting is required. This issue impacts the early identification of infectious and zoonotic factors and could result in the transmission of possible pathogens [23]. Evidence from other studies demonstrates that farmers and their veterinarians frequently neglect to report abortion cases despite the existence of mandatory reporting requirements, with such scenarios usually driven by self-interest and concerns about potential consequences, including farm isolation for suspected cases of Brucellosis or other notifiable diseases, in addition to financial and logistical challenges [33,34,35]. In addition, Hui people are largely concentrated in Ningxia and their lifestyle and dietary practices follow Islamic traditions [15]. We also found that some Hui populations do not report abortion cases because they are under religious influence to not profit from dead animals. Efforts have thus been made in attempt to increase the reporting of abortion cases by publicizing information on the transmission of Brucellosis and its main threats; despite these efforts, little progress has been made in some villages.

Secondly, animals that test negative for all *Brucella* spp. should be tested to determine whether other abortigenic pathogens, including *Campylobacter fetus* and *Leptospira* spp., are present in order to control ruminant abortions [35]. However, due to limited funding and laboratory capabilities, another limitation of this study is the fact that the aborted samples were not tested to determine if other pathogens were present.

Despite these limitations, this study provides valuable epidemiological information on Brucellosis in aborted livestock in Ningxia. The results could aid in the analysis of the spread and evolution of *Brucella* spp. in Ningxia and lead to novel vaccine development. They could provide enhanced insights into strain evolution and transmission pathways employing higher-resolution genomic markers (e.g., Whole-Genome Sequencing) in future studies.

## 5. Conclusions

In this study, *B. melitensis*, *B. abortus*, and *B. suis* isolated from aborted tissues of livestock from 2022 to 2023 in Ningxia were identified and analyzed using MLVA and MLST. Our research results suggest that comprehensive disinfection and safe treatment should be implemented after the abortion of livestock to interrupt transmission pathways to prevent livestock infection and thus human transmission. Moreover, aborted livestock should be examined to determine possible zoonotic causes, and targeted surveillance should be strengthened to improve the early detection of infectious causes, patricianly Brucellosis, which would be of benefit to the breeding industry and public health security. Such measures can be further developed by proposing prioritized monitoring of sheep flocks (primary hosts for *B. melitensis* ST8), particularly those experiencing abortions, and evaluating S2 vaccine efficacy in this context.

## Figures and Tables

**Figure 1 vetsci-12-00702-f001:**
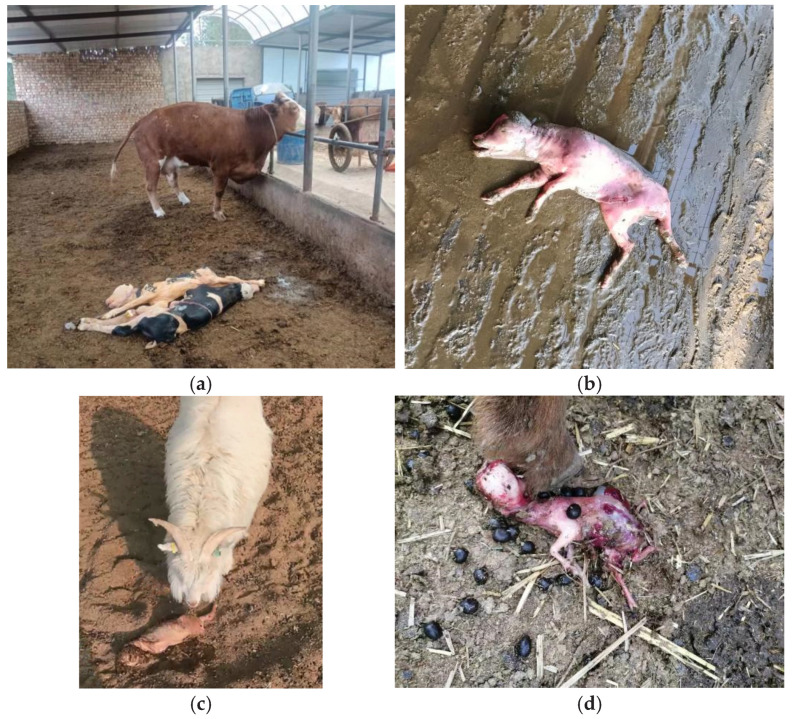
Aborted livestock used in this study. (**a**) Aborted beef cattle and stillbirths in Longde, Guyuan city, in June 2023. (**b**) Dairy cow stillbirth in Shapotou, Zhongwei city, in September 2023. (**c**) Aborted sheep and stillbirths in Yanchi, Wuzhong city, in 2022.12. (**d**) Sheep stillbirth in Xiji, Guyuan city, in March 2023. (We thank Xueyi Wang, Ruigang Wang, and Kuiju Zhang for their assistance in taking the photographs).

**Figure 2 vetsci-12-00702-f002:**
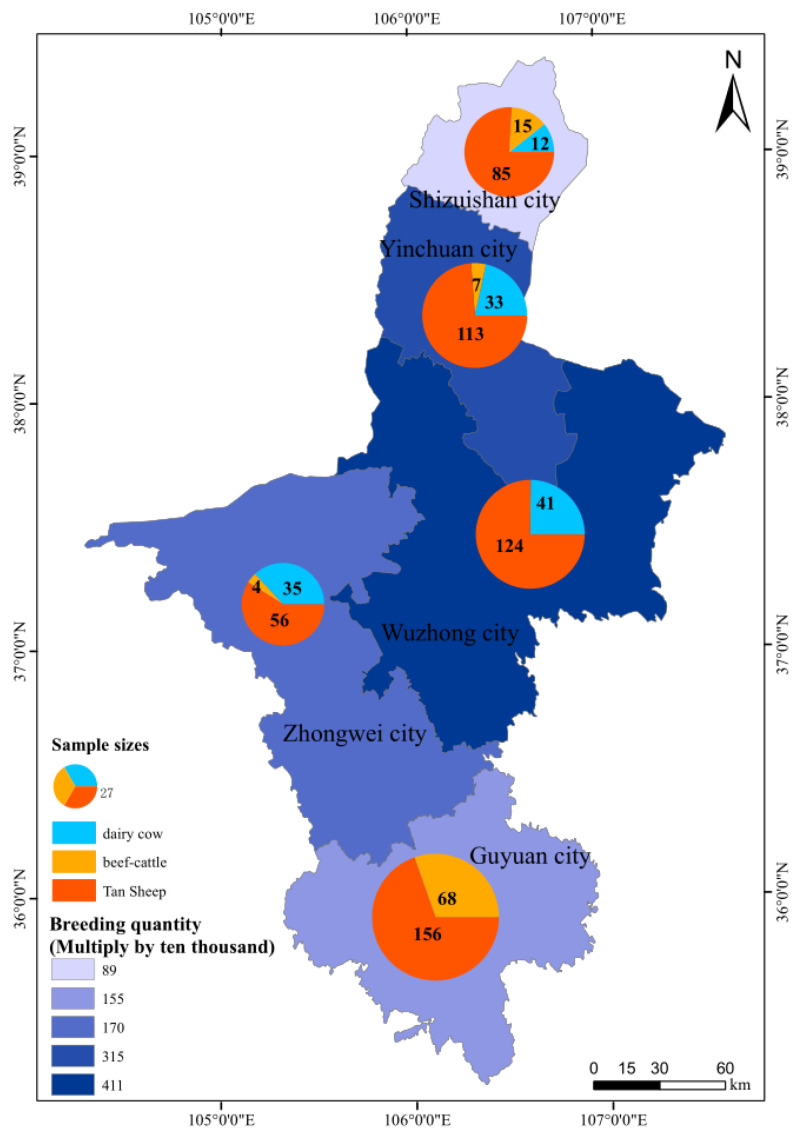
Sampling sizes of aborted livestock in this study and the number of breeding livestock in Ningxia.

**Figure 3 vetsci-12-00702-f003:**
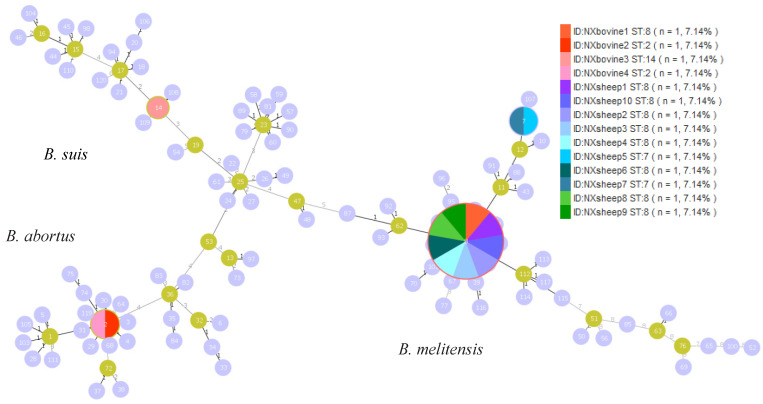
The minimum spanning tree of the 14 strains identified in this study was constructed using PHYLOZIV 2.0 software. Orange, red, light orange, and pink colors indicate strains isolated from cattle; other colors indicate strains isolated from sheep.

**Figure 4 vetsci-12-00702-f004:**
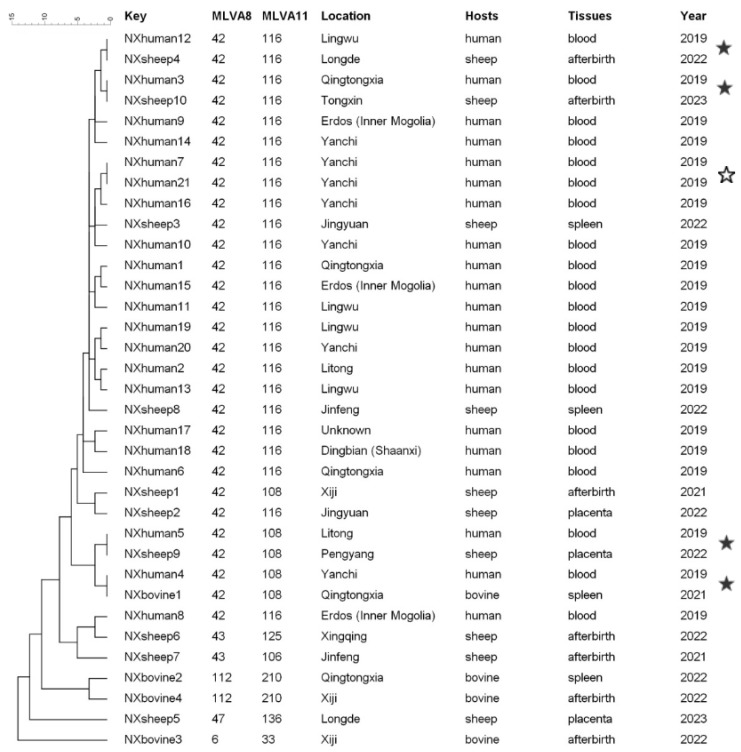
MLVA dendrogram of 14 strains isolated from aborted livestock in this study and 21 strains isolated from humans in the study by Liu et al. in Ningxia [15]. The hollow pentagram show shared genotypes in humans; the solid pentagram shows the shared genotypes in humans and livestock.

## Data Availability

All data are included in the manuscript.

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
