# Peer review of "Molecular Epidemiology of Brucella spp. in Aborted Livestock in the Ningxia Hui Autonomous Region, China"

_vetsci, 2025, doi:10.3390/vetsci12080702_

Round 1

Reviewer 1 Report

Comments and Suggestions for Authors

Comments on the manuscript vetsci-3696974 entitled

Molecular epidemiology of Brucella spp. in aborted livestock in Ningxia Hui Autonomous Region, China

This study presents the first systematic molecular epidemiological investigation of Brucella spp. in aborted livestock within Ningxia, China. The authors successfully isolated and characterized field strains using MLST and MLVA genotyping. The research design is robust, methodologies are appropriate (including BSL-3 protocols), data analysis is rigorous, and conclusions provide significant insights for brucellosis control in Ningxia and nationally. Results clearly demonstrate the co-circulation of multiple Brucella species (B. melitensis, B. abortus, B. suis) and provide molecular evidence of zoonotic transmission (shared MLVA genotypes). The manuscript is well-structured, and figures or tables are generally supportive.

Specific Comments:

  1. Novelty and Significance

Strengths: This work fills a critical knowledge gap on the molecular epidemiology of Brucella in aborted livestock in Ningxia. It reports the first isolation of B. suis in the region and provides compelling molecular evidence (8 shared MLVA genotypes) supporting livestock-to-human transmission, which is vital for targeted control. The relevance to Ningxia's "Three-Year Action Plan" enhances its practical impact.

Introduction: Further emphasize the specific epidemiological context and urgency in Ningxia. Integrate recent (<5 years) key references on the national distribution of prevalent STs (e.g., dominance of ST8 across China) and zoonotic transmission dynamics to better position this study's findings. Avoid redundancy with general background on economic impacts.

Discussion: Deepen the interpretation of the predominance of ST8 in Ningxia. Discuss whether ST8's prevalence in Northwest China reflects regional fitness advantages or virulence traits. Compare findings with Brucella strain distributions in neighboring provinces (e.g., Inner Mongolia, Gansu).

  1. Methods and Results

Sample Handling and Processing:

Page 2: Samples "stored at -20 °C until further processing."

Page 3 (DNA Extraction): "inactivated by heating at 100 °C... extracted from... tissues."

Clarification Needed: Specify whether DNA for qPCR was extracted directly from tissue samples post-storage or from bacterial cultures. The notable discrepancy between qPCR positivity (8.68%) and isolation rate (14/749 ≈ 1.87%) is appropriately attributed in the Discussion Page 9 to factors like residual DNA from dead bacteria. Recommend explicitly stating in Methods (Sections 2.2 and 2.4.1) the source of DNA for qPCR (tissues vs. cultures).

MLVA Analysis: Results Page 6 mention analysis of Panel 1 (MLVA-8), Panel 2A, and Panel 2B, yet report MLVA-8 and MLVA-11 genotypes. Clarify: Is MLVA-11 equivalent to Panel 1 + Panel 2A? The presentation of "6 MLVA-11 genotypes" requires explanation of its derivation.

Figure 4 (MLVA Dendrogram): This figure is crucial for demonstrating zoonotic transmission. Strongly recommend clearly marking the 8 pairs of shared genotype strains (e.g., NXhuman7 or NXhuman21) within the figure or legend using distinct symbols (e.g., ★) for immediate reader identification.

Figures 1: Figure 1 provides valuable field context. Enhance the caption by including approximate location (city or county) and timeframe (year or season) of photography.

  1. Discussion and Conclusions

Strengths: The discussion effectively links findings to Ningxia’s "Three-Year Action Plan" (28.5% decline in human cases in 2023), underscores the importance of disinfection and protective measures ( "mask and gloves initiative"), and thoughtfully addresses underreporting challenges influenced by cultural or religious factors (Hui population, Page 9).

Areas for Enhancement: ST8 Dominance: As noted, contextualize the ST8 predominance within broader regional or national Brucella strain distributions.

Refined Control Recommendations: The conclusion suggests "strengthening targeted surveillance." Elaborate by proposing prioritized monitoring of sheep flocks (primary hosts for B. melitensis ST8), particularly those experiencing abortions, and evaluating S2 vaccine efficacy in this context.

Limitations: While underreporting and undetected co-pathogens are well-discussed, consider adding: "Future studies employing higher-resolution genomic markers (e.g., Whole Genome Sequencing) could provide enhanced insights into strain evolution and transmission pathways."

It is suggested that the discussion be deleted. “Usually, the abortion rate is less than 2% in healthy herds (Menzies, 2011). Most abortions occur sporadically, and the rate is generally under 5% (Daniel Givens and Marley, 2008). Thus, a rate between 2% and 5% might suggest the presence of one or more infectious agents (Jonker et al., 2023; Menzies, 2011). In the studied region, there are nearly 10,000,000 livestock. Assuming that 10% of female cattle are retained and the abortion rate is only 1%, there would be nearly 10,000 cases of abortion. According to relevant regulations in Ningxia, China, any animal suspected of being infected with an epidemic disease, including cases of abortion shall be reported to the local competent veterinary department. Nevertheless.

  1. Formatting and Minor Issues

Terminology Consistency:

Add section numbers 2.2 and 2.3. The subsequent section numbers need to be re-sequenced.

In 2.4.3, unify "minutes" and "min".

Minor Issues should be pay attention.

References: Generally current and relevant. Verify that all in-text citations  match the reference list .

Author Statement Page 10: Correct "publication in One Health" to align with the journal title: "publication in Vet. Sci.".

Funding Statement: Appropriately detailed with grant numbers.

Author Response

Comments 1: “Further emphasize the specific epidemiological context and urgency in Ningxia. Integrate recent (<5 years) key references on the national distribution of prevalent STs (e.g., dominance of ST8 across China) and zoonotic transmission dynamics to better position this study's findings. Avoid redundancy with general background on economic impacts.”

Response 1: Thank you for pointing this out. We agree with this comment. Therefore, we have added the “B. melitensis ST8 is the most common species widely prevalent in 1990-2010” in line 76.

Comments 2: “Deepen the interpretation of the predominance of ST8 in Ningxia. Discuss whether ST8's prevalence in Northwest China reflects regional fitness advantages or virulence traits. Compare findings with Brucella strain distributions in neighboring provinces (e.g., Inner Mongolia, Gansu).”

Response 2: Thank you for pointing this out. We agree with this comment. Therefore, we have added the “B. melitensis ST8 is also the main species in Inner Mongolia, Xinjiang, Qinghai, Guangxi, being responsible for many cases of human Brucellosis.” in line 299.

Comments 3: “Clarification Needed: Specify whether DNA for qPCR was extracted directly from tissue samples post-storage or from bacterial cultures. The notable discrepancy between qPCR positivity (8.68%) and isolation rate (14/749 ≈ 1.87%) is appropriately attributed in the Discussion Page 9 to factors like residual DNA from dead bacteria. Recommend explicitly stating in Methods (Sections 2.2 and 2.4.1) the source of DNA for qPCR (tissues vs. cultures).”

Response 3: Thank you for pointing this out. We agree with this comment. Therefore, we have added the “The source of DNA for qPCR is tissues and the culture isolation. ” in line 129.

Comments 4: “MLVA Analysis: Results Page 6 mention analysis of Panel 1 (MLVA-8), Panel 2A, and Panel 2B, yet report MLVA-8 and MLVA-11 genotypes. Clarify: Is MLVA-11 equivalent to Panel 1 + Panel 2A? The presentation of "6 MLVA-11 genotypes" requires explanation of its derivation.”

Response 4: MLVA-11 is equivalent to Panel 1 + Panel 2A. 

Thank you for pointing this out. We agree with this comment. Therefore, we have added the “7 MLVA-11 genotypes (GT116, GT108, GT125, GT106, GT210, GT136, GT33)” in line 230.

Comments 5: “Figure 4 (MLVA Dendrogram): This figure is crucial for demonstrating zoonotic transmission. Strongly recommend clearly marking the 8 pairs of shared genotype strains (e.g., NXhuman7 or NXhuman21) within the figure or legend using distinct symbols (e.g., ★) for immediate reader identification.”

Response 5: Thank you for pointing this out. We agree with this comment. Therefore, we have added the “★’’ and “” in the fig.for marking the shared genotype strains.

Comments 6: “Figures 1: Figure 1 provides valuable field context. Enhance the caption by including approximate location (city or county) and timeframe (year or season) of photography.”

Response 6: Thank you for pointing this out. We agree with this comment. Therefore, we have added the location and timeframe of photography in line 193.

Comments 7: “Areas for Enhancement: ST8 Dominance: As noted, contextualize the ST8 predominance within broader regional or national Brucella strain distributions.”

Response 7: Thank you for pointing this out. We agree with this comment. Therefore, we have added the “The phenomenon that B. melitensis ST8's prevalence in Northwest China and even in southern China recently, may be reflects regional fitness advantages of the strain. It mainly resulted by the frequent transportation and circulation of live livestock in China mainland.” in line 302.

Comments 8: “Refined Control Recommendations: The conclusion suggests "strengthening targeted surveillance." Elaborate by proposing prioritized monitoring of sheep flocks (primary hosts for B. melitensis ST8), particularly those experiencing abortions, and evaluating S2 vaccine efficacy in this context.”

Response 8: Thank you for pointing this out. We agree with this comment. Therefore, we have added the “Elaborate by proposing prioritized monitoring of sheep flocks (primary hosts for B. melitensis ST8), particularly those experiencing abortions, and evaluating S2 vaccine efficacy in this context.” in line 352.

Comments 9: “Limitations: While underreporting and undetected co-pathogens are well-discussed, consider adding: "Future studies employing higher-resolution genomic markers (e.g., Whole Genome Sequencing) could provide enhanced insights into strain evolution and transmission pathways.”

Response 9: Thank you for pointing this out. We agree with this comment. Therefore, we have added the “ It could provide enhanced insights into strain evolution and transmission pathways employing higher-resolution genomic markers (e.g., Whole Genome Sequencing) in future studies.” in line 341.

Comments 10: “It is suggested that the discussion be deleted. “Usually, the abortion rate is less than 2% in healthy herds (Menzies, 2011). Most abortions occur sporadically, and the rate is generally under 5% (Daniel Givens and Marley, 2008). Thus, a rate between 2% and 5% might suggest the presence of one or more infectious agents (Jonker et al., 2023; Menzies, 2011). In the studied region, there are nearly 10,000,000 livestock. Assuming that 10% of female cattle are retained and the abortion rate is only 1%, there would be nearly 10,000 cases of abortion. According to relevant regulations in Ningxia, China, any animal suspected of being infected with an epidemic disease, including cases of abortion shall be reported to the local competent veterinary department. Nevertheless.””

Response 10: Thank you for pointing this out. We agree with this comment. Therefore, we have delete the sentences.

Comments 11: “Formatting and Minor Issues

Terminology Consistency:

Add section numbers 2.2 and 2.3. The subsequent section numbers need to be re-sequenced.

In 2.4.3, unify "minutes" and "min".

Minor Issues should be pay attention.

References: Generally current and relevant. Verify that all in-text citations  match the reference list .

Author Statement Page 10: Correct "publication in One Health" to align with the journal title: "publication in Vet. Sci.".

Funding Statement: Appropriately detailed with grant numbers.”

Response 11: Thank you for pointing this out. We agree with this comment. Therefore, we have added the section numbers 2.2 and 2.3. The subsequent section numbers had been re-sequenced. And unified the "minutes" and "min" to "minutes" and "minute". Corrected "publication in One Health" to align with the journal title: "publication in Vet. Sci.". in line 370. We have appropriately detailed with funding grant numbers.

Reviewer 2 Report

Comments and Suggestions for Authors

The manuscript entitled “Molecular Epidemiology of Brucella spp. in Aborted Livestock in Ningxia Hui Autonomous Region, China”  is of great interest because of the topic it deals with, since in several countries around the world the problem of brucellosis continues to be a topical issue and there appears to be an increasing need for biosecurity measures to be implemented, especially in certain types of animal husbandry.

General comments:

  • The manuscript appears well structured and organised in each section.
  • The introduction by the authors is very interesting and well-edited, but I would advise them to add information on the prevalence of brucellosis in the area if these data are available.
  • Generally, about the ‘materials and methods’ and ‘results’ sections, I congratulate the authors on the clear expositions of the methods and the easy accessibility of the results. The same can be partially said about the discussion of the results.

Specific considerations:

  • Among the other aetiological agents researched as potentially responsible for abortion, it would be interesting to know whether abortigenic salmonellae and agents of a viral and protozoan nature were also included. Furthermore, it would be interesting to have information on the respective prevalence in the area. The authors could use this information to better argue the case for what emerged from their investigation in the discussion section.

Author Response

Thank you very much!

Comments 1: “The introduction by the authors is very interesting and well-edited, but I would advise them to add information on the prevalence of brucellosis in the area if these data are available.”

Response 1: Thank you for pointing this out. We agree with this comment. Therefore, we have added "the prevalence of brucellosis of bovine in the area was 2.46%, and that of sheep was 3.85% in 2022." in line 82.

Comments 2: “Among the other aetiological agents researched as potentially responsible for abortion, it would be interesting to know whether abortigenic salmonellae and agents of a viral and protozoan nature were also included. Furthermore, it would be interesting to have information on the respective prevalence in the area. The authors could use this information to better argue the case for what emerged from their investigation in the discussion section.”

Response 2: Thank you very much for pointing this out. We agree with this comment. But due to the limited funding and laboratory capabilities, the aborted samples were not tested for other pathogens in this study. We will investigated for other zoonotic causes of aborted livestock if the funding is OK in the future.

Best wishes!